# SpatialComposer: 3D Spatial Object Insertion via Image Gaussian Composition

## Abstract

With the rapid advancement of open-world image generation models in recent years, a series of image editing tasks have achieved excellent performance. However, considering object insertion as a representative example, this task still presents three primary challenges. First, the inserted object should maintain identity consistency with the reference object while preserving the original scene in non-edited regions. Second, the spatial position and scale of the inserted object should be reasonable and align with user expectations. Third, the inserted object should harmonize with other image components, typically involving object style and surface illumination harmonization. To address these challenges, we propose SpatialComposer, which leverages depth-aware image Gaussians to construct a spatially-structured scene representation from a single scene image and models object insertion as Gaussian composition, thereby achieving effective preservation of scene and object identity while enabling precise control over the scale and 3D spatial position of the inserted object. Subsequently, based on pre-trained diffusion generative models, we introduce a simple yet effective refinement method for the object harmonization process. By designating only the Gaussian components corresponding to the inserted object as trainable parameters, SpatialComposer avoids unintended modifications to other regions while simultaneously addressing both object-scene integration and scene detail preservation. Furthermore, recognizing that current object insertion benchmarks lack consideration for depth-aware position control, we construct a specialized benchmark featuring high-resolution scene images with substantial depth complexity. Comprehensive evaluations demonstrate that SpatialComposer achieves comparable or superior performance over state-of-the-art object insertion approaches across all three aforementioned challenges.

## 1 Introduction

Object insertion is a fundamental image editing task involving seamless integration of specified objects into target scene locations. Recent diffusion-based models (e.g., Stable Diffusion (Rombach et al., 2022b), FLUX (Labs, 2024)) have enabled significant evolution beyond previous methods (Tripathi et al., 2019; Zhang et al., 2020; Liu et al., 2021; Niu et al., 2022; Cong et al., 2020; Ling et al., 2021; Sofiiuk et al., 2021; Cong et al., 2022).

Current methods fall into two categories: *training-based methods*(Song et al., 2025; Wang et al., 2025; Chen et al., 2024b) that fine-tune pre-trained inpainting models conditioned on reference objects, and *training-free methods*(Chen et al., 2024c; Wang et al., 2024) that manipulate intermediate features and attention during inference. These methods enable intuitive personalized customization of real or synthesized images using arbitrary references, providing accessible automated tools for creating and modifying visual content.

Despite promising results, object insertion faces three main challenges: *1). Object and scene consistency*. The forward diffusion process randomness and encoder-decoder downsampling information loss make maintaining detail consistency challenging for current methods. *2). 3D Spatial and scale controllability*. Existing approaches control insertion through object masks or bounding boxes with optional text prompts, specifying only two-dimensional positioning and failing to accurately control depth-related spatial position and scale. Our experiments show that even with depth text prompts,

previous methods often fail to meet expectations. *3). Style and illumination harmony.* Current methods struggle with style consistency, natural lighting, and color harmonization needed for seamless visual integration.

In this paper, we propose a novel depth-aware object insertion method based on Gaussian Kerbl et al. (2023) representation, which we term SpatialComposer. In order to tackle the above three challenges, our approach contains three main steps: Gaussian fitting, Gaussian composition, and Object refinement. In the first step, by leveraging a pre-trained monocular depth estimation network, combined with a back-projection Gaussian initialization strategy, we efficiently construct consistent object and scene Gaussian representations with meaningful spatial structure. In the second step, object insertion is implemented through the composition of object Gaussians with scene Gaussians. Through user-specified scaling and translation operations applied to the object Gaussians, our approach achieves precise control over both the depth and scale of the inserted object. In the third step, following the composition of object and scene representations, we introduce a simple yet effective refinement method for inserted objects based on pre-trained inpainting diffusion models and illumination harmonization models. With separable scene Gaussians and object Gaussians, we designate only the object Gaussian components as trainable parameters, thereby preventing unintended modifications to other scene regions. This design circumvents the limitations of pre-trained diffusion generative models regarding resolution constraints and reconstruction fidelity. It also preserves the integrity of unmodified scene areas while harnessing the supervisory signals provided by pre-trained diffusion models for harmonizing inserted objects with the scene.

The existing TF-ICON benchmark (Lu et al., 2023b) for image-guided object insertion consists of generated $512 \times 512$ scene images with simple spatial structures, where insertions lack spatial relationships with scene components. This benchmark is insufficient for our evaluation. To fill this research gap, we collect a novel benchmark dataset comprising over 200 high-resolution scenes with complex structures and over 200 objects spanning 9+ categories including animals, vegetables, food, people, and vehicles. Scenes include over 100 real photographs and over 100 artistic images across multiple styles: oil painting, cartoon, anime, watercolor, and pixel art. We constructed over 200 semantically coherent insertion cases. We name the dataset Depth-Aware Object Insertion (DAOI) Dataset and will release it soon.

Ablation experiments demonstrate the effectiveness of our image Gaussian representation and initialization method. Using a single image and relative depth estimates, we rapidly construct scene Gaussians with meaningful depth while maintaining high-quality reconstruction. Leveraging Gaussian compositionality, scaling and translating object Gaussians enables precise control of scale and placement at any location. The proposed refinement module exhibits strong generalization and excellent performance without additional fine-tuning. Unlike existing methods, our approach handles ultra-high-resolution images while preserving fine details and demonstrates robust performance across diverse artistic styles, confirming practical applicability.

In summary, our main contributions include: (1) We propose a depth-aware image Gaussian representation and a back-projection initialize strategy to enhance the performance of Gaussian fitting while providing meaningful depth information. (2) To the best of our knowledge, we are the first to consider the precise control of 3D spatial position and scale for inserted objects in object insertion tasks. Based on our proposed representation, we provide a solution to this problem. This representation also achieves stronger preservation of non-edited scene regions and object identity. (3) We collect a high-resolution object insertion benchmark featuring complex scene spatial structures, providing a high-quality evaluation platform for advancing research in this domain. (4) Experimental results demonstrate the superior performance and robustness of SpatialComposer across diverse stylistic and real scenes.

## 2 RELATED WORK

### 2.1 TRAINING-BASED OBJECT INSERTION

Built upon pre-trained text-to-image or inpainting models, training-based object insertion methods employ techniques such as LoRA (Hu et al., 2022) and Adapters (Houlsby et al., 2019) to fine-tune the models using paired data, i.e., scene images with and without specific objects. These approaches typically introduce new network structures to accept the scene image, reference object, and spa-

tial mask as conditions. Several works (Yang et al., 2023; Song et al., 2022; Chen et al., 2024a; Canet Tarrés et al., 2024; Yuan et al., 2024; Kulal et al., 2023; Zhang et al., 2023a; Song et al., 2024; He et al., 2024) extract reference features via visual encoders and trainable modules, influencing UNet through cross-attention, summation, or custom fusion. Some methods (Zhang et al., 2023b; Chen et al., 2024b) use composite images with pasted objects as conditions. Insert Anything and UniCombine (Song et al., 2025; Wang et al., 2025) adopt Diffusion Transformer (DiT) for generation control. For environmental harmony, Zerocomp (Zhang et al., 2025) conditions on depth, surface normal, albedo, and shading. Objectmate (Winter et al., 2024b) uses $2 \times 2$ grids as input to produce coherent insertion results. Anydoor (Chen et al., 2024b) combines ID extractors with high-pass filters for dual feature control. These methods require large paired datasets and have insufficient generalization ability. To avoid large dataset dependency, DreamCom and DreamEdit(Lu et al., 2023a; Li et al., 2023) fine-tune embeddings using DreamBooth(Ruiz et al., 2023) with few images. DreamEdit(Li et al., 2023) uses DDIM Inversion(Mokady et al., 2023), while DreamCom(Lu et al., 2023a) employs masked attention control. OmniPaint(Yu et al., 2025) trains separate insertion/removal models with cycle consistency loss. To reduce paired data, ObjectDrop (Winter et al., 2024a) trains removal models on small datasets, then collects large synthetic datasets for insertion training. Overall, when there exists a significant gap between training and inference data, the preservation of non-edited scene regions and object identity, as well as the harmony between objects and scenes, exhibit substantial degradation. These methods are also unable to control the depth-related spatial positioning of object insertion. In contrast, SpatialComposer is training-free, enables precise control over 3D spatial positioning, and has been validated to deliver stable performance across real-world scenes and a variety of stylistic scenes.

## 2.2 TRAINING-FREE OBJECT INSERTION

Instead of parameter updating, training-free object insertion methods manipulate the intermediate features or attention maps during the inference stage. TF-ICON (Lu et al., 2023b) uses three-branch inference with DDIM-inverted noise from scene and reference images, along with composite noise placing resized object noise into target regions. Scene and object features guide cross-attention in the composite branch. Similarly,(Li et al., 2024) employs three-branch inference with self-attention fusion for object preservation and text-guided modification. FreeCompose(Chen et al., 2024c) iteratively optimizes editing through key-value replacement and DDS-loss (Hertz et al., 2023). Prime-Composer (Wang et al., 2024) introduces a correlation diffuser computing cross-attention between target and reference features in UNet self-attention layers, with attention maps injected during denoising for appearance preservation alongside region-constrained cross-attention. Constrained by the reconstruction and generation capabilities of the underlying foundation models, these methods exhibit limitations in preserving non-edited regions and maintaining object identity. Likewise, they do not provide effective control over the spatial placement of inserted objects.

## 3 METHOD

The pipeline of SpatialComposer is illustrated in Fig. 1. It comprises three main components: Gaussian fitting, Gaussian composition and object refinement. We begin by introducing the foundational concepts of 3D Gaussians in Sec. 3.1, and we also introduce the latent diffusion model in Appendix A. Subsequently, we provide detailed descriptions of our proposed Depth-Aware Image Gaussian (DA-ImgGS) representation, its initialization, fitting and composition in Sec. 3.2 and the Diffusion-Based Object Refinement method in Sec. 3.3.

### 3.1 3D GAUSSIAN SPLATTING (3DGS)

Our Gaussian representation builds upon standard 3D Gaussians with task-specific adaptations. Hence, we first introduce the foundational concepts of 3DGS in this section. 3DGS is an explicit scene representation using 3D Gaussians Kerbl et al. (2023), with each Gaussian $\mathcal{G}$ defined by mean $\boldsymbol{\mu} \in \mathbb{R}^3$ and covariance matrix $\Sigma \in \mathbb{R}^{3 \times 3}$:

$$\mathcal{G}(\mathbf{x}) = e^{-\frac{1}{2}(\mathbf{x}-\boldsymbol{\mu})^T \Sigma^{-1}(\mathbf{x}-\boldsymbol{\mu})}. \tag{1}$$

Additionally, each Gaussian includes spherical harmonics (SH) $\mathbf{c} \in \mathbb{R}^k$ for color and opacity $\alpha \in \mathbb{R}$. The covariance matrix decomposes as $\Sigma = RSS^T R^T$, where $R \in \mathbb{R}^{3 \times 3}$ is the rotation matrix and

Figure 1: We first initialize and fit both scene and object images using our proposed depth-aware image Gaussian representation. The object is then scaled and positioned at the desired location through scaling and translation operations to generate the composed Gaussians. Subsequently, we employ a refinement method based on pre-trained diffusion models to optimize the object Gaussians, thereby achieving harmonization between the object and the scene.

$S = \text{diag}([s_x, s_y, s_z])$ is the diagonal scale matrix. The rotation matrix $R$ is constructed from quaternion $\mathbf{v} = [r_w, r_x, r_y, r_z]$. For rendering, 3D Gaussians are projected onto pixel coordinates given camera pose $W$, with pixel-space covariance matrix defined as:

$$\Sigma_{pix} = JW\Sigma W^T J^T, \tag{2}$$

where $J$ is the Jacobian matrix of the affine approximation of the projection transformation. The color at each pixel can then be obtained through alpha-blending of $N$ overlapping Gaussians at that pixel in depth order:

$$\mathbf{c}_{pix} = \sum_i^N \mathbf{c}_i \alpha_i \prod_j^{i-1} (1 - \alpha_j), \tag{3}$$

where $\mathbf{c}_i$ and $\alpha_i$ represent the color and density of each Gaussian at that pixel, weighted by the covariance matrix $\Sigma$. The differentiable rendering process enables end-to-end optimization of all Gaussian parameters based on image reconstruction loss.

### 3.2 DEPTH-AWARE IMAGE GAUSSIAN (DA-IMGGS)

We modify the parameter configuration, initialization, and optimization of standard 3D Gaussian to develop depth-aware image Gaussian. This adaptation accommodates our application requirements, enhances fitting performance on single images, while endowing the scene representation with usable spatial structure.

Standard 3DGS defines Gaussian means in world coordinates as 3D positions. During rendering, these world coordinates are transformed to camera coordinates through the camera's extrinsic and intrinsic matrices before being projected onto the image plane, enabling image rendering from arbitrary viewpoints. However, for applications requiring rendering only from a fixed camera pose, we define our depth-aware image Gaussian directly in Normalized Device Coordinates (NDC). NDC represents a standardized coordinate system in computer graphics where coordinates are normalized to the range $[-1, 1]$. This coordinate system enables graphics rendering to adapt seamlessly across display devices with different resolutions and aspect ratios.

The covariance matrix in standard 3DGS represents an ellipsoid in 3D space, which projects to ellipses on different image planes during rendering. Since image Gaussians only require rendering onto a fixed image plane, we reduce the covariance matrix to 2D, where $\Sigma_{2d} \in \mathbb{R}^{2 \times 2}$ represents an elliptical Gaussian distribution parallel to the fixed image plane. In standard 3DGS, object colors observed from different viewpoints depend on multiple factors including viewing angle, lighting conditions, and material properties. Higher-order spherical harmonics decompose this lighting and material information into coefficient sets, providing efficient and smooth representations of complex lighting effects such as shadows, specular highlights, and viewpoint-dependent color variations. However, since image Gaussian rendering operates from a fixed viewpoint without 3D scene lighting variations, simple RGB values sufficiently capture the required surface colors. This reduction in Gaussian representation parameters improves scene fitting quality. Finally, the parameters of our

depth-aware image Gaussian include: mean $\mathbf{u} \in \mathbb{R}^3$, where the 2D covariance matrix is decomposed via Cholesky decomposition as $\Sigma_{2d} = L \cdot L^T$, requiring storage of only three elements from the lower triangular matrix $L$, denoted as $L = (l_{11}, l_{21}, l_{22})^T \in \mathbb{R}^3$, opacity $\alpha \in \mathbb{R}$, and RGB color representation $\mathbf{c} \in \mathbb{R}^3$.

When supervised with only a single image and its corresponding monocular depth estimation, random initialization of Gaussians followed by optimization produces floating Gaussian components in mid-air. These components disrupt the correct spatial structure of the scene and lead to poor fitting performance. Therefore, we adopt a back-projection initialization method based on the input image and depth estimation. The mean of each initial Gaussian component in NDC is determined using the normalized pixel coordinates and depth estimation of the corresponding pixel. Color values are initialized directly from the pixel RGB values. The initial axis lengths of the ellipse corresponding to the covariance matrix are computed based on the scene image resolution, focal length, and pixel depth values by analyzing the distribution range of initial Gaussian components in NDC space. This back-projection initialization approach enables rapid fitting of image Gaussians while preventing the emergence of floating Gaussian components. The supervision during optimization combines multiple loss terms: L1 and SSIM losses between the rendered and source scene images, regularization terms for occupancy and covariance, and L1 loss with respect to the depth estimation values:

$$\mathcal{L}_{gs} = (1 - \lambda_{ssim})\mathcal{L}_1 + \lambda_{ssim}\mathcal{L}_{ssim} + \mathcal{L}_{1\_depth} + \lambda_{reg\_\alpha}\mathcal{L}_{reg\_\alpha} + \lambda_{reg\_\Sigma}\mathcal{L}_{reg\_\Sigma}, \quad (4)$$

we set $\lambda_{ssim}$ to 0.2, $\lambda_{reg\_\alpha}$ and $\lambda_{reg\_\Sigma}$ to 0.01. After fitting Gaussians to the scene and object images, the object can be placed at any spatial location within the scene Gaussians at arbitrary scale by scaling and translating the object Gaussians. This process can be interactively accomplished by users in real-time within the visualization interface. Details are provided in Appendix D.

### 3.3 DIFFUSION-BASED OBJECT REFINEMENT

After achieving depth-aware object insertion through Gaussian combination of scene and object, the inserted object requires further refinement to harmonize its style and surface illumination with the surrounding environment. We propose a simple yet effective object refinement approach that leverages a pretrained diffusion model to harmonize the rendered image. Diffusion models pretrained on large-scale datasets inherently capture diverse output distributions encompassing various visual styles and lighting conditions. For cross-domain object insertion tasks (real object insert to stylized scene), we directly exploit the knowledge embedded in the pretrained FLUX.1-Fill-dev model to refine the inserted object, thereby achieving style harmonization between the object and scene.

We denote the combined Gaussians of scene and object as $\mathcal{G}com$, where $R$ represents the differentiable rendering process. Based on the combined Gaussians, we can simultaneously render the composite image and the opacity of object Gaussians to obtain the object mask: $\mathbf{I}_{com}, \mathbf{M}_{obj} = R(\mathcal{G}_{com})$. We introduce an coefficient $s \in [0, 1]$ to control the refinement strength. When $s = 0$, the method outputs the original image without modification, while $s = 1$ performs complete inpainting of the masked region. Let $T$ denote the total number of inference steps in the diffusion model, corresponding to $T$ different noise levels. Given refinement strength $s$, the corresponding noise level is computed as $T' = \lceil sT \rceil$. Let $\mathcal{E}$ denote the encoder of the diffusion model. The latent variable corresponding to the composite image is $\mathbf{z}_{com} = \mathcal{E}(\mathbf{I}_{com})$. The initial latent variable for the inference process is then defined as:

$$\mathbf{z}_{T'} = \sigma(T')\boldsymbol{\epsilon} + (1 - \sigma(T'))\mathbf{z}_{com}, \boldsymbol{\epsilon} \sim \mathcal{N}(0, \boldsymbol{I}). \quad (5)$$

DiT models such as FLUX process inputs by dividing them into patches and converting the values within each patch into tokens. Let the size of each patch be $k = h \times w$, with a total of $N$ patches. We apply similar processing to $\mathbf{M}_{obj}$, transforming it into a patch-wise binary mask $\mathbf{M}_{patch} \in \{0, 1\}^N$:

$$m_n = \left[\sum_{i=1}^{k} \mathbf{M}_{obj\_n,i} > 0\right] \in \{0, 1\}. \quad (6)$$

We denote the diffusion model used for refinement as $G_{refine}$, which follows the flow matching framework. At each time step $t$ during the refinement process, we first perform a single denoising step on the noisy latent variables.

$$\mathbf{z}_{t-1} = \mathbf{z}_t - \Delta t \cdot G_{refine}(\mathbf{z}_t, t|c), \quad (7)$$

where $c$ represents the textual condition for the model. In our implementation, we employ a simple template format `"XXX style of a XXX"` that provides basic descriptions of object categories and styles. Subsequently, we perform an overwrite operation on the latent variables based on the patch-wise mask.

$$\mathbf{z}_{t-1} = \mathbf{M}_{patch} \odot \mathbf{z}_{t-1} + (1 - \mathbf{M}_{patch}) \odot \mathbf{z}_{ref}(t-1), \tag{8}$$

with

$$\mathbf{z}_{\text{ref}}(t) = \begin{cases} \sigma(t)\boldsymbol{\epsilon} + (1 - \sigma(t))\mathbf{z}_{com} & t > 0, \\ \mathbf{z}_{com} & t = 0. \end{cases} \tag{9}$$

This ensures precise reconstruction of the original scene image in regions outside the object mask.

In real-scene object insertion, object refinement focuses primarily on achieving consistency between object surface lighting and the surrounding environment. Since widely-used open-source text-to-image diffusion models and inpainting models such as Stable Diffusion or FLUX possess limited knowledge in lighting harmonization, we additionally leverage the pre-trained illumination harmonization model LBM (Chadebec et al., 2025), denoted as $G_{light}$. For object refinement in real-scene, we first apply low-strength refinement to harmonize the inserted object, then perform additional lighting harmonization on the refined image using $G_{light}$. The refinement process for real-scene domain object insertion can be expressed as:

$$\mathbf{I}_{harmony} = G_{light}(\text{Refinement}(\mathbf{I}_{com}, s)). \tag{10}$$

After completing the refinement process on the rendered image following object insertion, the harmonized image from the diffusion model can directly provide supervision for Gaussian parameter updates due to the differentiable nature of the rendering process. However, during the refinement process, diffusion models often introduce unwanted modifications or detail loss in non-target regions due to constraints in their generative capabilities and supported resolutions. Therefore, we set only the object Gaussians as learnable to avoid affecting the scene Gaussians.

## 4 EXPERIMENTS

### 4.1 EXPERIMENT SETUP

**Data Preparation** Due to the lack of high-quality object insertion datasets with complex spatial relationships and cross-domain scenarios, we construct a high-resolution dataset containing diverse real photographs and artistic scenes with complex spatial structures. For scenes, we select high-quality photographs from the DIODE dataset (Vasiljevic et al., 2019), high-resolution artworks from WikiArt (Saleh & Elgammal, 2024), frames from anime and cartoon videos, and generate additional scenes using GPT-Image-1 (OpenAI, 2024) and FLUX-pro-1.1-ultra. Stylized scenes encompass anime, cartoon, comic, pixel art, watercolor, and over ten oil painting styles from WikiArt. Real-world scenes include diverse indoor/outdoor environments, various lighting conditions (strong, weak, nighttime), and different light sources (natural, artificial, colored). This yields 225 high-quality scenes across multiple domains. For objects, we collect web resources and use SAM (Kirillov et al., 2023) to segment objects from COCO (Lin et al., 2014), obtaining 255 objects spanning over 9 categories including animals, vegetables, food, people, and vehicles. We provide over 200 semantically coherent scene-object combinations for evaluation. All experiments are conducted on this dataset. We also report results on the TF-ICON benchmark in Sec 4.5.

**Experiment Details** For Gaussian fitting, we employ the Adam optimizer with learning rates of $0.01$ for means, $2.5e-3$ for opacities, $0.01$ for RGB values, and $0.1$ for covariances. We fix the global random seed to 42. Since our method involves only forward inference of the refinement model and Gaussian fitting, a single A6000 GPU is sufficient to support object refinement using FLUX.1-Fill-dev.

**Baselines** To validate our method's effectiveness, we compare against both training-based and training-free object insertion approaches. The training-based baselines include InsertAnything, Uni-Combine, and AnyDoor, while the training-free baselines comprise Freecompose, Primecomposer.

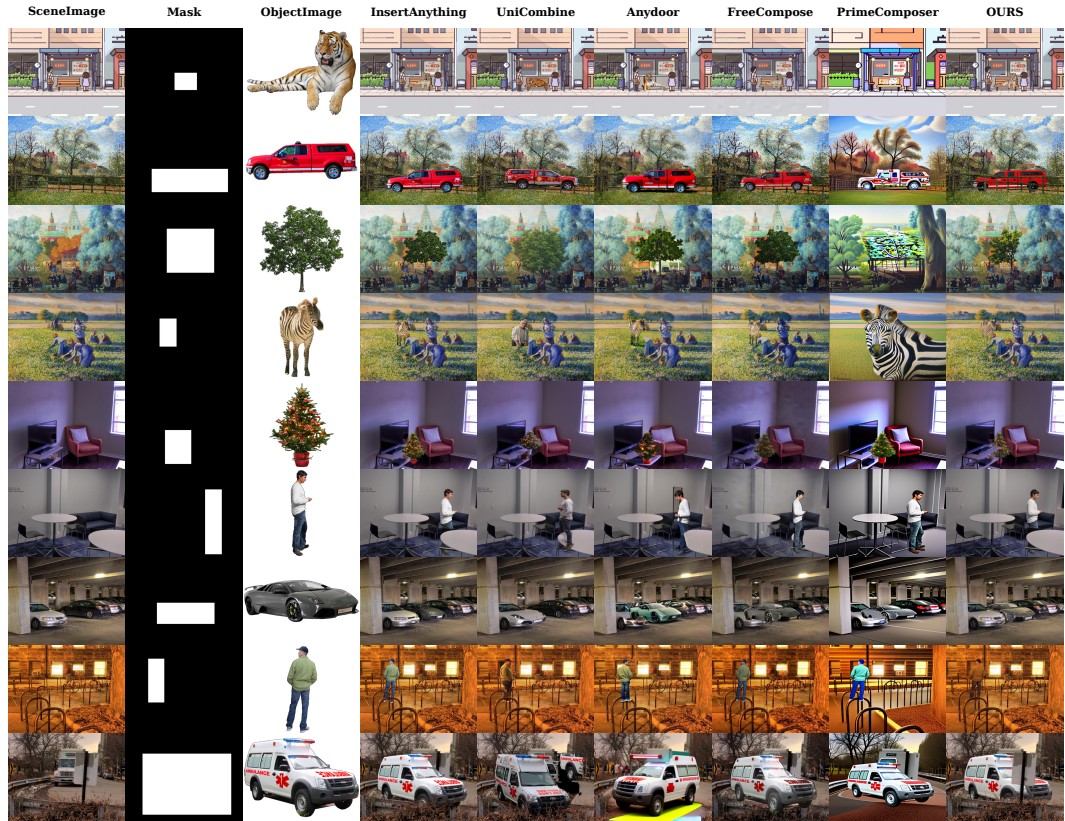

Figure 2: Visual comparison results between our SpatialComposer and baselines. Zoom in to observe details.

## 4.2 QUANTITATIVE EVALUATION

Existing metrics measure non-edited region preservation by calculating distance between low-level features in generated and original scenes. We compute PSNR between these regions and the original scene. For object identity, prior works (Lu et al., 2023b; Wang et al., 2024; Chen et al., 2024c) measure distance between high-level semantic or low-level perceptual features from edited regions and reference objects. Our method considers depth-related positioning during insertion. After handling occlusion relationships, feature distance between edited regions and references actually increases. When domain gaps exist between scenes and objects, color and texture modifications are required for seamless integration, degrading quantitative metrics. We therefore exclude these measures. Follow prior works, we adopt the cosine similarity between the CLIP image embedding of the edited region in the result image and the CLIP text embedding of the corresponding text prompt for the edited region (formatted as `"XXX style / professional photograph of a XXX"`) to comprehensively measure both inserted object style and category. CLIP-based similarity metrics do not account for alignment with human visual perception, often yielding results inconsistent with human subjective evaluation. Therefore, we introduce VQAScore (Lin et al., 2024), a metric specifically designed for generated image assessment that demonstrates superior alignment with human subjective judgment, to evaluate both the overall visual quality of inserted objects and their surrounding regions, as well as their alignment with text prompts (also formatted as `"XXX style / professional photograph of a XXX"`). However, spatial positioning and scale reasonableness remains difficult to assess automatically.

Given the lack of automatic metrics aligned with human perception for interactive generation tasks, this field relies on user studies for evaluation. We recruited 40 participants to evaluate SpatialComposer and baselines across four dimensions: overall quality, object-environment harmony, spatial positioning and scale reasonableness, and consistency with reference objects and original scenes.

Table 1: Quantitative comparison of different methods on different benchmarks. **Bold** indicates the best result, underline denotes the second best, and $^{\dagger}$ marks training-free methods.

| Method | Our DAOI Dataset | | | TF-ICON Dataset | | |
|---|---|---|---|---|---|---|
| | $\text{PSNR}_{bg}$ | CLIPScore | VQAScore | $\text{PSNR}_{bg}$ | CLIPScore | VQAScore |
| AnyDoor | 31.52 | 0.287 | 0.520 | 23.74 | 0.279 | 0.493 |
| UniCombine | 32.03 | 0.255 | 0.447 | 26.61 | 0.286 | 0.567 |
| InsertAnything | 36.82 | 0.296 | 0.535 | 31.91 | 0.285 | 0.557 |
| Freecompose$^{\dagger}$ | 23.56 | 0.281 | 0.529 | 21.33 | 0.283 | 0.546 |
| Primecomposer$^{\dagger}$ | 16.82 | 0.268 | 0.445 | 12.32 | 0.272 | 0.590 |
| SpatialComposer$^{\dagger}$ | **46.99** | **0.299** | **0.552** | **33.14** | **0.307** | **0.653** |

Table 2: User study on our dataset across different evaluation dimensions.

| | AnyDoor | UniComb | InsertAnything | Freecomp$^{\dagger}$ | Primecomp$^{\dagger}$ | SpatialComp$^{\dagger}$ |
|---|---|---|---|---|---|---|
| **Qual** | 2.94 | 4.96 | 8.92 | 1.75 | 0.55 | **80.88** |
| **Harm** | 2.11 | 6.71 | 7.17 | 1.84 | 0.74 | **81.43** |
| **Reas** | 2.30 | 4.32 | 7.08 | 1.38 | 0.74 | **84.19** |
| **Cons** | 2.21 | 5.61 | 9.38 | 1.75 | 0.46 | **80.61** |

Participants identified the best method per dimension, and we computed average vote percentages across all dimensions. Table. 2 shows SpatialComposer achieves comparable or superior performance in automatic metrics. Our dataset comprises high-resolution real photographs and artworks with complex spatial structures and diverse styles at insertion locations. These characteristics challenge existing methods. SpatialComposer overcomes these through depth-aware Gaussian representations and effective use of pre-trained diffusion models, achieving significantly superior user study performance.

### 4.3 QUALITATIVE EVALUATION

As shown in Fig. 2, InsertAnything demonstrates the best performance among the baselines, showing some understanding of object depth relationships and displaying reasonable depth relations in certain insertion results. However, its capabilities for style harmonization in cross-domain insertion and illumination processing in real scenes remain limited. UniCombine exhibits stronger style harmonization capabilities for cross-domain object insertion but frequently generates results with significant errors or unreasonable outcomes. AnyDoor shows limitations in object-scene harmony and the visual quality in edited regions. FreeCompose lacks understanding of depth information and demonstrates limited harmonization performance. Beyond the issues encountered by other methods, PrimeComposer also exhibits significant problems in preserving the original scene image. Overall, SpatialComposer achieves comparable or superior performance in terms of image quality, reasonableness of spatial positioning and scale, consistency of object identity with non-edited regions of the scene, and harmonization between the scene and objects. More results are provided in Appendix B. Due to file size constraints, we provide vector-format versions of the visualization comparison figures in the supplementary material.

### 4.4 ABLATION STUDY

**Gaussian representation and initialization method** We simplify standard 3D Gaussians to better adapt them to our task and employ a back-projection-based initialization method. This approach not only significantly enhances scene fitting quality but also achieves superior spatial structure representation. As shown in Fig. 3, our proposed Gaussian representation and initialization method substantially improve the fitting quality of scene reconstruction. We also visualize the Gaussian means under different configurations, demonstrating that our proposed Gaussian settings and initialization method yield the most reasonable spatial structure, which serves as the foundation for subsequent object insertion operations.

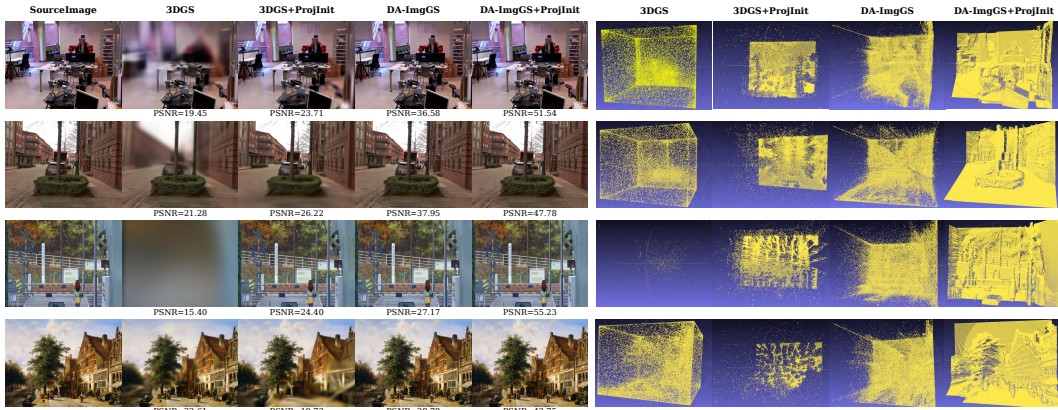

Figure 3: Comparison of fitting performance under different Gaussian representations and initialization strategies. Both the proposed depth-aware image Gaussian and initialization strategy demonstrate improved fitting quality and spatial structure.

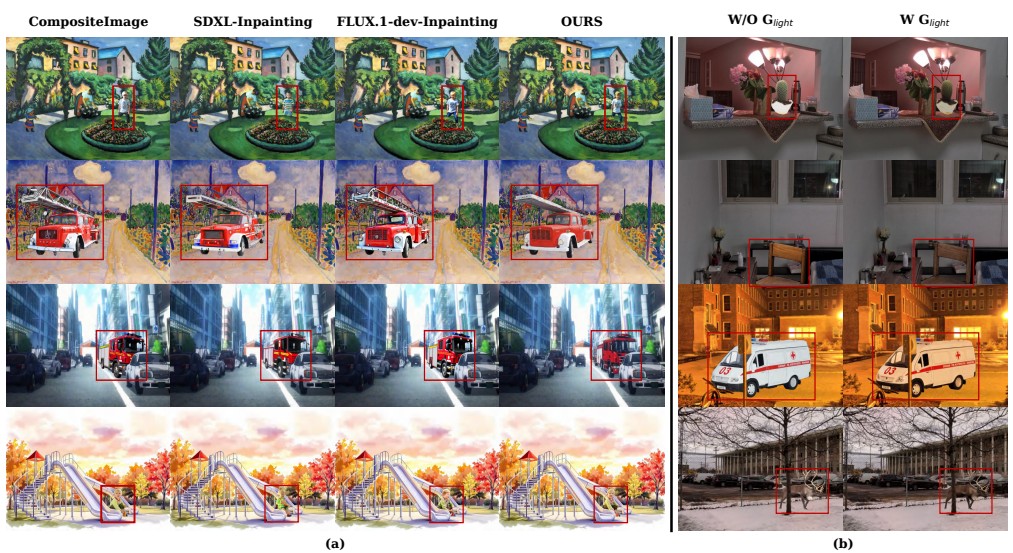

Figure 4: (a) demonstrates the compatibility of our pipeline with different pre-trained diffusion models and the effectiveness of our refinement method; (b) shows that incorporating pre-trained illumination harmonization models in real-scene object insertion enhances the harmony between object surface lighting and the environment.

**Refinement Foundation Model** In the refinement processing of inserted objects, our pipeline is compatible with different foundation models. Fig. 4 (a) compares the refinement results across different foundation models. Our proposed refinement method achieves the best overall performance in maintaining object identity while achieving style consistency between objects and scenes.

**Illumination Harmonization Model in Real Scenes** In Fig. 4 (b), we validate the limitations of pre-trained text-to-image and inpainting models when addressing object surface illumination harmonization in real scenes, as well as the necessity of incorporating pretrained illumination harmonization models. The illumination harmonization model better captures environmental lighting information, including intensity, color, and direction.

**Refinement Strength** Finally, we validate the impact of the strength coefficient $s$ on the results during the refinement process. As shown in Fig. 5, higher refinement strength produces results that are more harmonious with the scene style but simultaneously weakens the preservation of object identity. An appropriate value should be determined based on the degree of domain gap between

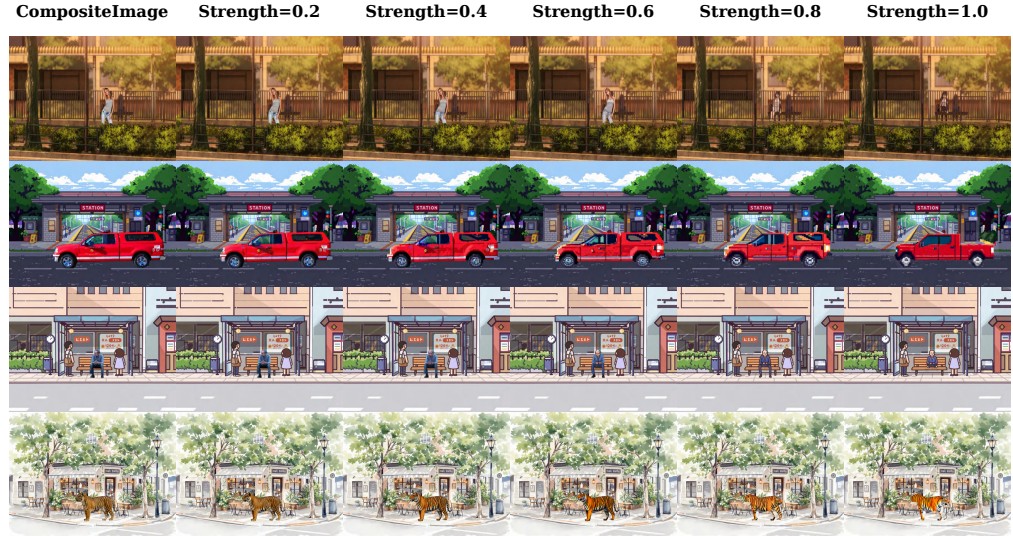

Figure 5: The impact of different object refinement intensities on the results.

the inserted object and the scene. In our experiments, we select $s = 0.6$ as the default setting for stylized scene object insertion and $s = 0.2$ for real scene object insertion.

### 4.5 RESULTS ON TF-ICON BENCHMARK

Prior works most closely related to ours, TF-ICON and Primecomposer, primarily employ the TF-ICON Benchmark. This dataset comprises 95 stylized scene object insertion cases and 237 real scene object insertion cases, constructed from approximately 30 scene images and about 100 object images. Due to its low scene resolution, synthetic scene generation, and simplistic object insertion scenarios, this dataset cannot fully satisfy the requirements of our task. Although the object insertion locations in this dataset do not involve complex spatial relationships, SpatialComposer still achieves performance comparable to or superior to existing methods in terms of overall quality. The quantitative results presented in Table.1 further support our observations. We also present a visual comparison of different methods on the TF-ICON benchmark in Appendix Fig.6.

## 5 CONCLUSION AND LIMITATION

In this paper, we introduce SpatialComposer, which effectively reconstructs high-fidelity scene representations with meaningful depth structure, enabling precise control over object scale and 3D spatial positioning during insertion. To address insertion disharmony, we employ a refinement method based on pretrained diffusion models, achieving reasonable harmonization that aligns object and lighting with the surrounding scene. Furthermore, we constructed the DAOI dataset, in which we collected over 200 high-resolution scene images with complex spatial structures and diverse styles, along with over 200 multi-category object images. Experiments demonstrate our method's superiority, achieving comparable or superior performance across multiple evaluation dimensions.

In our task setting, given only a single image of the insertion object, we cannot construct a complete Gaussian representation of the object and can only model the visible surface shown in the single image. Consequently, in the subsequent Gaussian composition operations, our method only supports scaling and translation of the object Gaussians, but does not support rotation of the object Gaussians. Reconstructing individual object Gaussians from single object images represents an actively pursued research direction in the 3D Gaussian field. With the rapid advancement of techniques in this domain, incorporating object 3D Gaussian reconstruction models based on a single object image may provide an effective approach to address the limitations of our method.

## ETHICS STATEMENT

This work adheres to the ICLR Code of Ethics. Our research focuses on image editing tasks, where all data are sourced from open-source datasets, generated through model API calls, and publicly available internet data, with no involvement of personal privacy information. All data used in this study are publicly available and were used in accordance with their respective licenses and terms of use. We have properly cited all data sources and respected the original creators' rights. While our method demonstrates improvements in object insertion, we recognize that like other AI technologies, it could potentially be misused to generate unsafe visual content. We encourage responsible deployment and further research into safety mechanisms. We believe this work contributes positively to the research community and poses minimal ethical concerns when used responsibly.

## REPRODUCIBILITY STATEMENT

We have made every effort to ensure the reproducibility of our work. Our constructed dataset DAOI will be publicly available alongside our framework and experimental source code upon publication, while TF-ICON serves as a public benchmark. Our proposed method is detailed in Sec. 3, which includes the hyperparameters involved in the loss terms. Additional experimental details such as optimizers, learning rates, and random seeds are provided in Sec. 4. We commit to making the complete framework and experimental source code, as well as the constructed benchmark, publicly available as soon as possible after publication.

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

## A  LATENT DIFFUSION MODEL

In this work, we leverage pre-trained text-conditioned latent diffusion models (Rombach et al., 2022a) to guide the object refinement process. These models operate in a learned latent space through an encoder-decoder architecture, where $\mathcal{E}(\cdot)$ and $\mathcal{D}(\cdot)$ represent the encoder and decoder components, respectively. The diffusion process involves forward noise addition followed by reverse denoising operations within this latent representation. Given an input image $\mathbf{x}_0$, we first encode it into the latent space as $\mathbf{z}_0 = \mathcal{E}(\mathbf{x}_0)$. During training, this latent representation is progressively corrupted through the forward diffusion process, transforming $\mathbf{z}_0$ into $\mathbf{z}_t$:

$$\mathbf{z}_t = \sqrt{\bar{\alpha}_t}\mathbf{z}_0 + \sqrt{1 - \bar{\alpha}_t}\boldsymbol{\epsilon}, \boldsymbol{\epsilon} \sim \mathcal{N}(0, \mathbf{I}), \tag{11}$$

for $t \in [1, T]$, where $\bar{\alpha}_t = \prod_{s=1}^{t} 1 - \beta_s$, and $\beta_s$ represents the variance schedule at timestep $s$. Subsequently, a denoising U-Net is trained to predict the added noise conditioned on $c$ using the following objective function:

$$\mathcal{L} = \mathbb{E}_{\mathcal{E}(\mathbf{x}_0), \boldsymbol{\epsilon} \sim \mathcal{N}(0, \mathbf{I}), t} \left[ \|\boldsymbol{\epsilon} - \boldsymbol{\epsilon}_\theta(\mathbf{z}_t, t, c)\|_2^2 \right], \tag{12}$$

where $\boldsymbol{\epsilon}_\theta$ represents the denoising U-Net.

## B  ADDITIONAL VISUAL COMPARISON RESULTS

In Figs. 7 and 8, we present additional visualized comparisons of generation results.

## C  MULTI-OBJECT INSERTION EXAMPLES

SpatialComposer supports inserting multiple objects in a single scene. Since the Gaussians of these objects and the scene Gaussians are distinguishable, it avoids the accumulation of influence on other Gaussians when inserting multiple objects. In Fig. 9, we show some results of multiple object insertion in scenes.

## D  INTERACTIVE GAUSSIAN COMPOSITION BASED ON VISUALIZATION

In the scene-object composition phase, we leverage the point clouds of Gaussian means from both the scene and object, which are preserved during the Gaussian fitting process, and visualize these two point clouds in a unified coordinate system using the Open3D library. The system supports 360-degree rotation of the entire coordinate system via mouse interaction, and allows real-time adjustment of the object Gaussian's scale, x-coordinate, y-coordinate, and z-coordinate through four pairs of keyboard keys, with continuous tracking and output of the scale and coordinate parameters during the adjustment process. Upon completion of the adjustment and closure of the visualization window, the scale and coordinate parameters are passed to the corresponding functions to perform operations on the object Gaussian, which is then composed with the scene Gaussian. As illustrated in Fig. 10, through this visualization-based Gaussian composition approach, we can precisely control the scale and position of the object Gaussian, enabling us to place the chair behind two pillars at different depths within the scene.

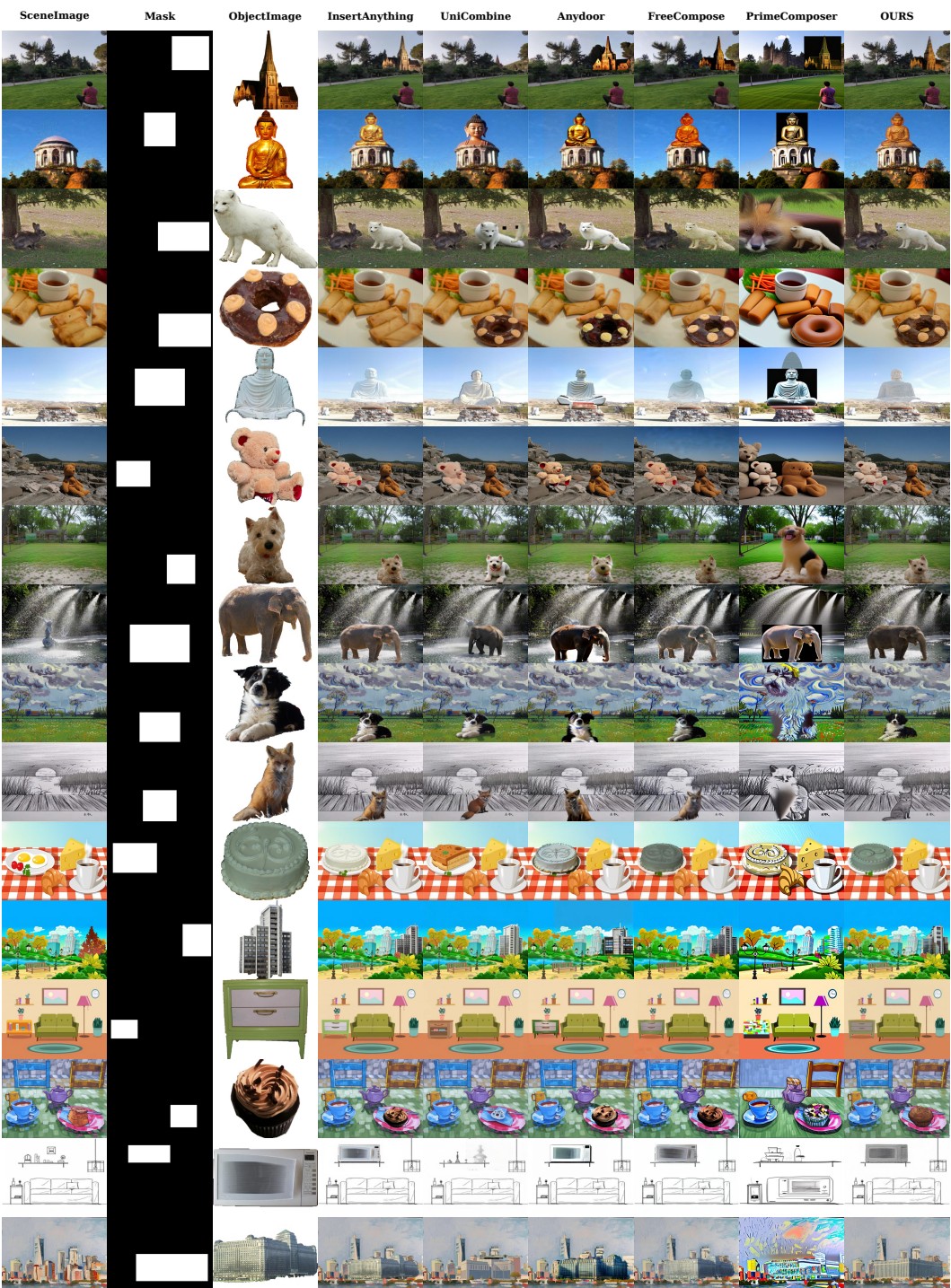

Figure 6: Visualization of comparative results on TF-ICON benchmark, zoomed in for detailed observation.

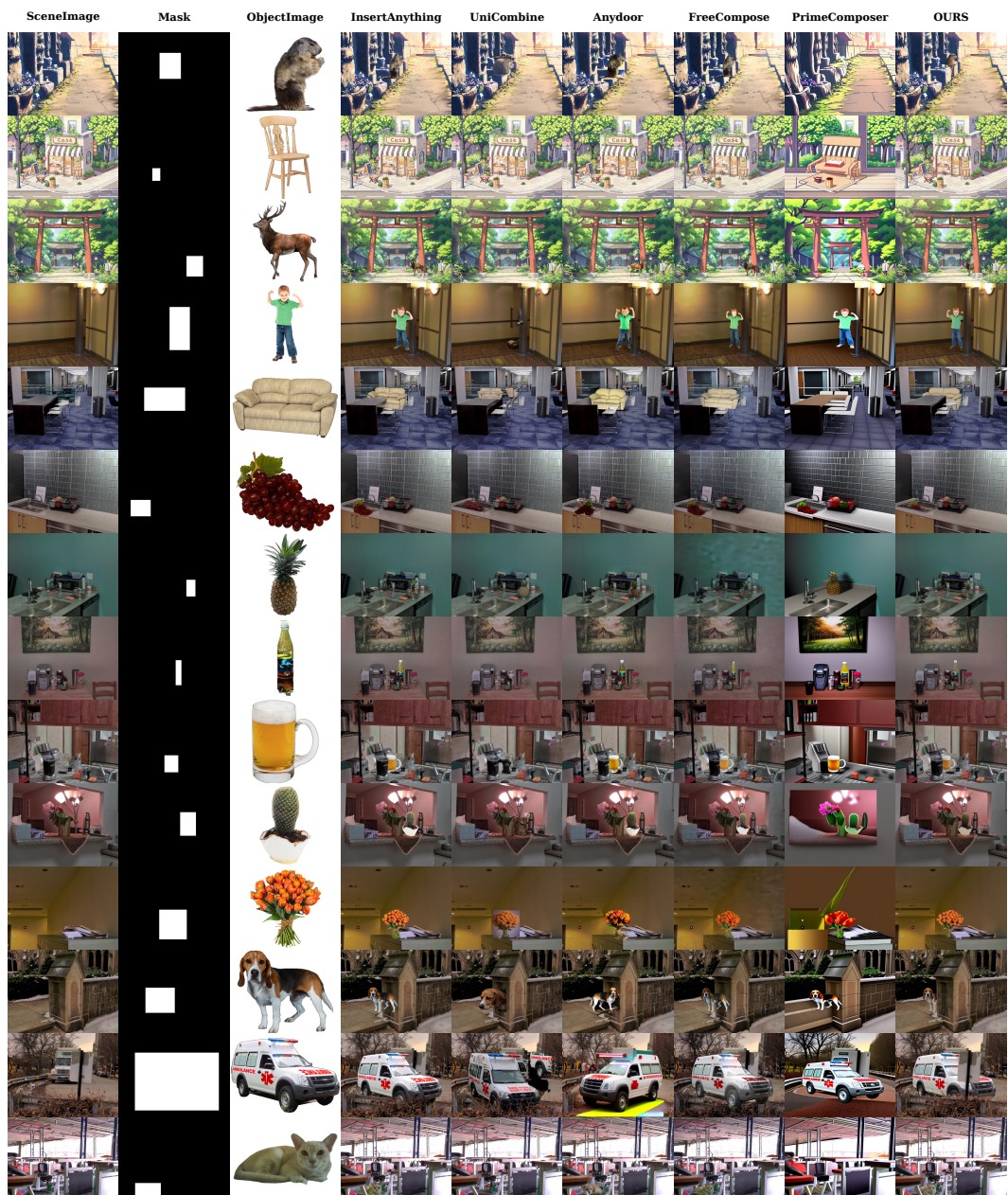

Figure 7: Visualization of comparative results, zoomed in for detailed observation.

## E    THE USE OF LARGE LANGUAGE MODELS

During the writing process of this paper, we utilized large language models to enhance the manuscript quality, including employing large language models to correct grammatical errors and modify wording and expressions to achieve a more formal and academic style.

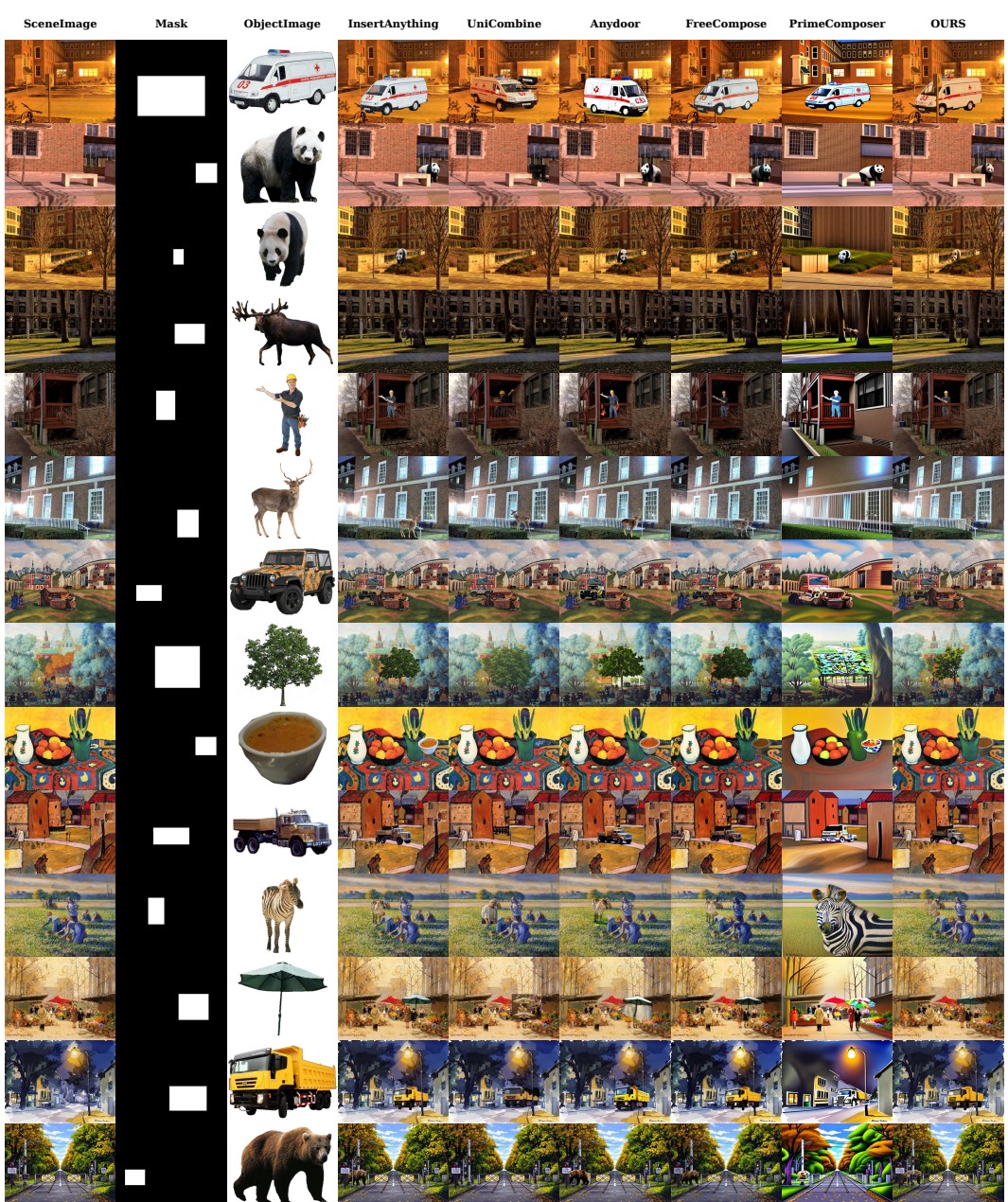

Figure 8: Visualization of comparative results, zoomed in for detailed observation.

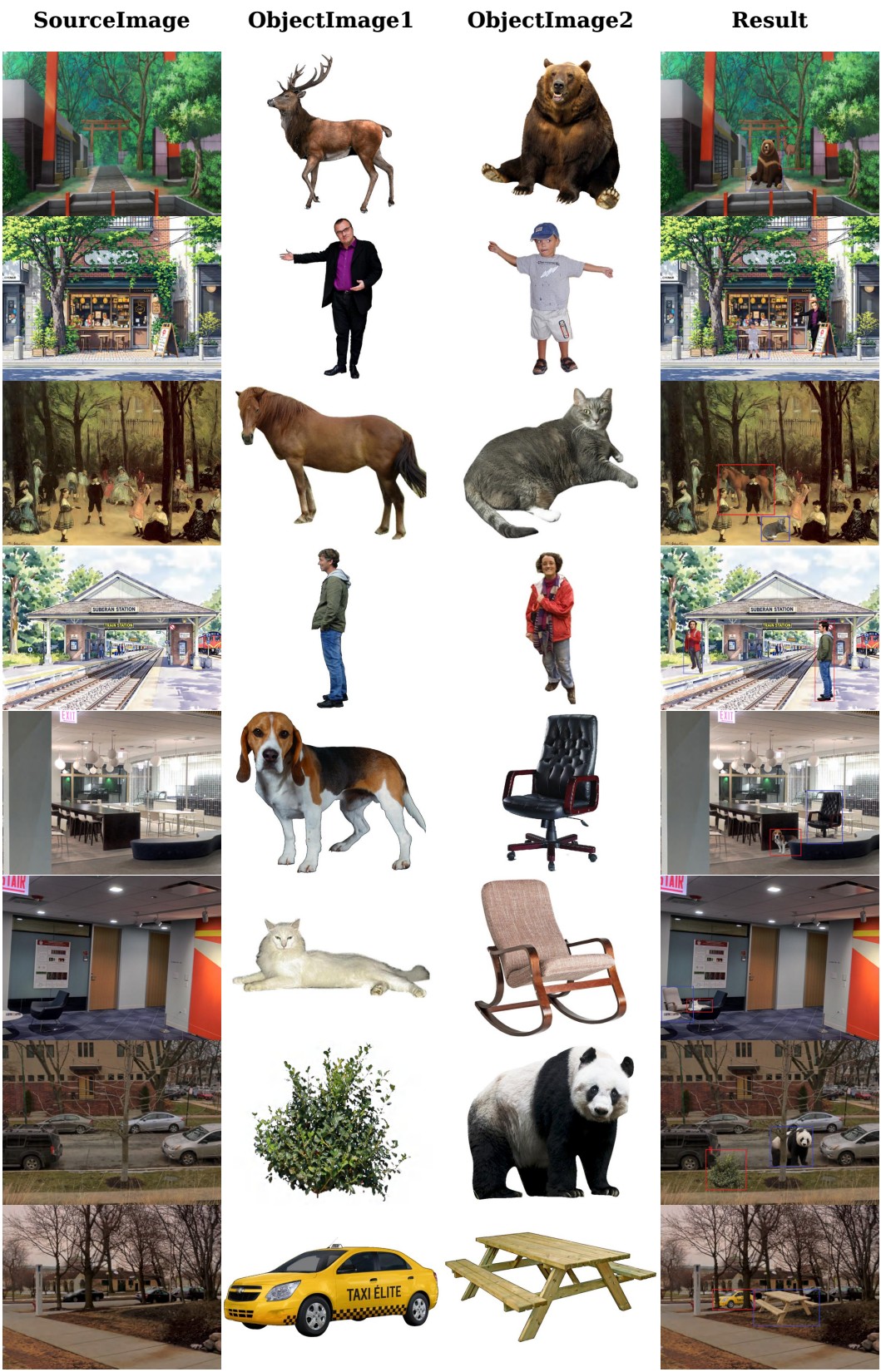

Figure 9: SpatialComposer supports multi-object insertion within a single scene while avoiding the adverse effects of multiple object insertion operations on other regions of the image, zoomed in for detailed observation.

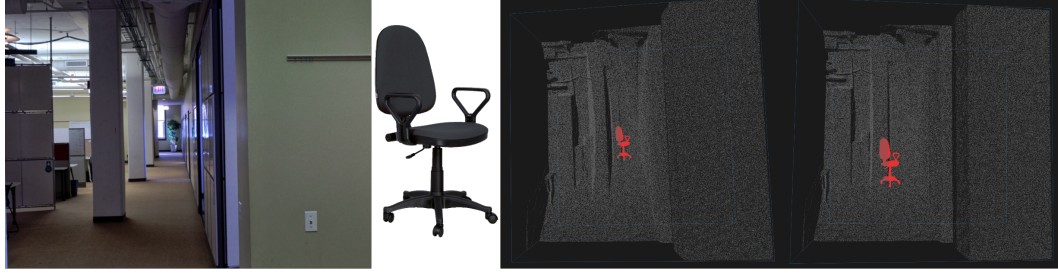

Figure 10: The first image on the left shows the scene, the second depicts the object to be inserted into the scene, and the third and fourth images demonstrate that through the visualization-based scene-object Gaussian composition process, we can precisely control the placement of the object Gaussian behind the first pillar and the second pillar in the scene, respectively.

