# OpenReview forum: "SpatialComposer: 3D Spatial Object Insertion via Image Gaussian Composition"
_ICLR.cc/2026/Conference — Submitted to ICLR 2026_

### Official Review · Reviewer_TPis · 2025-10-22

**Soundness:** 2
**Presentation:** 1
**Contribution:** 2
**Rating:** 4
**Confidence:** 2

**Summary:**

This paper tackles the object insertion task—adding a reference object into an existing scene image while preserving (1) object identity, (2) spatial realism (position, scale, depth), and (3) visual coherence with the scene (lighting, color, style).
The authors propose SpatialComposer, a method that constructs a depth-aware image Gaussian representation of the scene and formulates object insertion as Gaussian composition in 3D space. Only the Gaussian components corresponding to the inserted object are optimized (a “training-free” refinement step) on top of a pretrained diffusion model to ensure seamless blending.
A new benchmark dataset featuring scenes with higher depth complexity and image resolution is also introduced. Experiments demonstrate that SpatialComposer achieves competitive or superior results compared to existing approaches in terms of identity preservation, spatial control, and visual blending.

**Strengths:**

Clear problem definition (identity + spatial + style coherence). Methodologically coherent combination of Gaussian representation and diffusion-based refinement. Practical relevance for image editing, AR, and creative applications. New benchmark and comprehensive experiments add value.

**Weaknesses:**

1. Limited analysis of robustness and generalization to challenging real-world scenes (occlusion, reflection, lighting).
2. High pipeline complexity (depth estimation + Gaussian modeling + diffusion blending).
3. Dataset coverage may be narrow; unclear whether it generalizes to real “in-the-wild” data.
4. Innovation level is moderate; mainly a well-executed combination of existing ideas.
5. Missing deeper discussion on limitations and sensitivity analysis.
6. Some of the images in the paper do not use vector graphics and are very blurry when enlarged.

**Questions:**

1. How computationally expensive is the full pipeline? What is the average runtime or GPU memory usage for a single insertion?
2. Is the method practical for interactive or real-time use?
3. How sensitive is the method to errors in depth estimation or Gaussian fitting?

---

> ### Author Response · Authors · 2025-11-19
>
> Dear Reviewer TPis,
>
> We sincerely thank you for your diligent review work and for providing valuable questions and suggestions for our work.
>
> First, we apologize for the confusion. We noticed you used quotation marks around "train-free" and emphasized the word "optimize" in our harmonization process description. We need to clarify this terminology.
>
> In our understanding based on related works in the image generation/editing field [1][2], "train-free" primarily distinguishes methods from "train-based" approaches that collect task-specific data and fine-tune base generative models' parameters. Train-free methods do not require collecting target task domain data to fine-tune open-source pretrained generative models, but instead directly utilize model inference results or manipulate the inference process to achieve downstream task applications. The term "train-free" refers specifically to the base generative model, not implying the entire method involves no optimization process whatsoever. We apologize that this terminology caused misunderstanding.
>
> [1] Jiang R, Fu X, Zheng G, et al. Energy-guided optimization for personalized image editing with pretrained text-to-image diffusion models[C]//Proceedings of the AAAI Conference on Artificial Intelligence. 2025, 39(4): 4048-4056.
>
> [2] Chang J, Kim J, Ye J C. Training-Free Reward-Guided Image Editing via Trajectory Optimal Control[J]. arXiv preprint arXiv:2509.25845, 2025.
>
> **Response to Weakness 1:**
>
> Overall, solving occlusion problems is essentially precise depth control of inserted objects, which we are the first to address in object insertion tasks and represents our core contribution. We considered illumination issues and introduced the LBM model due to limitations in base text-to-image models. However, due to existing model limitations on illumination harmonization tasks, we cannot handle highly complex lighting effects like reflections—this is primarily due to available pretrained model capabilities.
>
> Our dataset selects numerous challenging cases with complex spatial structures from real data (real-world photographs/authentic artistic images) to address occlusion problems. Experiments validate our method's significant advantage in handling occlusion (i.e., depth-dimension controllability in object insertion). Training-free methods directly leverage base model capabilities for new tasks without task-specific fine-tuning, so their effectiveness is bounded by base model capabilities. Illumination and reflection are challenging real-world physics-related problems in image/video generation/editing. Base text-to-image models, learning purely data-driven from training data, struggle to truly understand underlying real-world physics and show limited capability on these issues. Therefore, due to limited performance of our base text-to-image model on these two problems, we additionally employ the LBM model fine-tuned for illumination harmonization when processing real-world scene lighting—this was a necessary design choice after observing the base model's inability to adjust target region lighting based on global illumination. Synthesizing corresponding physical effects like reflections for objects in images is also challenging in image generation/editing, typically requiring fine-tuning base models on paired image data. For challenging scenarios with complex physics like reflections, our pipeline cannot yet handle them due to available pretrained model limitations.
>
> **Response to Weakness 2:**
>
> For depth estimation, we only need to utilize existing tools or depth information as part of input preparation/data preparation. Under our Gaussian representation design, scene/object Gaussian modeling is a fast, low-memory process. Once scene and object fitting is completed, these explicit representations can be directly stored as assets and loaded for subsequent merging and harmonization when needed. Object Gaussian harmonization does not involve model training or fine-tuning—we only modify the model inference process and leverage the differentiability of Gaussian rendering to harmonize object Gaussians end-to-end using inference results. In our response to your Question 1, we provide detailed statistics on runtime and memory consumption for the entire pipeline, along with performance comparisons to baseline methods using the same base models as reference.

---

> > ### Author Response · Authors · 2025-11-19
> >
> > **Response to Weakness 3:**
> >
> > First, we clarify that our dataset is for evaluating and testing method performance. Our method does not involve model fine-tuning, so the dataset does not affect the method's generalization capability. In dataset construction, a small portion of artistic scene images are generated, while most come from authentic artworks, and all real scene images are actual photographs.
> >
> > Due to individual researcher limitations, constructing a large-scale dataset is challenging. However, we carefully considered data coverage during construction and collected data spanning diverse scenarios to validate our method's performance and generalization. For object images, we collected at least 9+ major categories including people, animals, vehicles, and furniture. For stylized scene images, we collected data covering various artistic styles including paintings from different schools, anime, comics, watercolor, pixel art, and sketch art. For real scene images, our data includes indoor and outdoor settings, different time periods (daytime, dusk, and night), and different light sources (natural light, artificial light, and colored light sources) (this point should be emphasized in the main text). Within our capacity as individual researchers, we made every effort to maximize coverage. Performance on this dataset validates our method's general applicability across these diverse scenarios.
> >
> > **Response to Weakness 4:**
> >
> > To the best of our knowledge, we are the first to address precise 3D spatial position and scale control for inserted objects in object insertion tasks and provide an effective solution based on our representation design. We also contribute a small-scale but high-quality benchmark for evaluating model performance on this task.
> >
> > Our method provides a scene image representation that can be constructed from a single image with high reconstruction accuracy and supports extensions for attributes like depth and semantics. Leveraging this representation and existing generative model capabilities, we provide a potential application on downstream tasks without additional model training, achieving comparable or superior results to baseline methods and solving a practical problem. The pipeline construction for object insertion and precise control of spatial position and scale extensively utilize properties enabled by our proposed representation—issues not addressed by previous works.
> >
> > Regarding object harmonization, the limited technical approach space is inherent to training-free methods. Training-free methods require neither fine-tuning on task-specific datasets nor introducing additional modules, which constrains technical approach options. Most training-free image/video editing methods manipulate latent features during inference, including weighted fusion, complete/partial replacement, and attention-based fusion. For harmonization, we similarly design an intensity and region-controllable approach that leverages pretrained image generation models to adjust low-level information (style, texture, color) based on global image context while preserving object identity. Additionally, our refinement design is motivated by the observation that diffusion models form high-level semantic information like layout in early denoising stages, while focusing on low-level details like texture and color in final steps—aspects more relevant to harmonization factors like style and illumination. Therefore, by applying appropriate timestep noising to images with disharmonious objects and manipulating latents during subsequent denoising, we achieve harmonization while preserving identity.

---

> > > ### Author Response · Authors · 2025-11-19
> > >
> > > **Response to Weakness 5:**
> > >
> > > In our collected data, we considered as many different scenario variations as possible. In experiments on these data, we did not encounter significant performance degradation or unusability on any particular samples. Testing on our benchmark demonstrates that our method exhibits stronger robustness and capability in handling data from different distributions compared to baseline methods.
> > >
> > > Complex surface illumination control for inserted objects and certain complex physical phenomena are issues our method cannot currently solve, nor are they the core problems we aim to address in this work. These limitations stem primarily from performance constraints in related image generation or editing methods and represent challenging open problems in the field.
> > >
> > > Existing illumination harmonization models cannot handle certain levels of surface lighting complexity. Since our method does not involve fine-tuning existing models, illumination harmonization relies on base model capabilities. If available models are insufficient to model certain complex real-world physics during harmonization, our method cannot handle these complex issues either.
> > >
> > > Beyond these, our method requires a viable depth estimation method or relatively reliable depth values as input. However, in testing on scenes from our benchmark and the TF-ICON benchmark, the existing depth prediction model DepthAnythingV2 provides usable relative depth estimates for both stylized and real scenes.
> > >
> > > **Response to Weakness 6:**
> > >
> > > We apologize for the inconvenience caused by image quality issues. We have already addressed this at the end of Section 4.3 (Qualitative Evaluation) in the main text. Due to paper submission file size limitations, we cannot use vector graphics for all images. Therefore, we included an explanation in the paper and uploaded a full vector graphics version of all result figures in the supplementary materials.
> > >
> > > **Response to Question 1:**
> > >
> > > In our experimental environment (A6000 Ada), fitting scene Gaussians at 1024×1024 resolution takes 1 minute with memory usage under 2GB, while fitting object Gaussians at 512×512 resolution takes approximately 30 seconds with memory usage under 1GB. Scene and object Gaussian fitting can be fully parallelized. Additionally, fitting can be performed in advance—once completed, Gaussians can be saved and reused, skipping the fitting step in subsequent uses.
> > >
> > > The harmonization process includes approximately 10 seconds of diffusion model inference, where peak memory usage occurs. With FluxFill as the base model, peak memory is approximately 35GB due to FluxFill's large parameter count. (For reference, InsertAnything, which produced suboptimal results in our experiments, and Unicombine, which also uses Flux as fundamental model, have peak memory consumption of 44GB and 35GB respectively.) For limited memory scenarios, StableDiffusionXL-inpainting can be used as the base model, controlling memory within 12GB. Our ablation study also demonstrates results based on SDXL-inpainting. Subsequent object Gaussian harmonization completes within 1 minute.
> > > Therefore, excluding user interaction for specifying object Gaussian scale and 3D spatial position, the entire pipeline completes in 2-3 minutes. With pre-fitted and stored Gaussians, the process takes just over 1 minute. (For reference, InsertAnything, which achieved suboptimal results in our experiments, takes approximately 2 minutes to complete one object insertion at the same 1024×1024 resolution. Moreover, training-based methods require collecting paired task-domain data for model training and fine-tuning.)
> > >
> > > **Response to Question 2:**
> > >
> > > When using SDXL-inpainting as the base model, memory usage is within 12GB, making it runnable on consumer-grade GPUs. In our experimental environment (A6000 Ada), the complete process takes 2-3 minutes. If scene and object Gaussians are pre-fitted and saved, it takes just over 1 minute to complete.

---

> > > > ### Author Response · Authors · 2025-11-19
> > > >
> > > > **Response to Question 3:**
> > > >
> > > > Our method relies on available depth estimates. For example, if depth prediction cannot distinguish between two objects positioned one behind the other in the scene, it naturally cannot support inserting an object between them. However, in testing on scenes from our benchmark and the TF-ICON benchmark, the existing depth prediction model DepthAnythingV2 provides usable relative depth estimates for both stylized and real scenes, and we have not encountered cases where depth estimates are insufficient to support subsequent operations.
> > > >
> > > > For Gaussian fitting of depth values, our proposed back-projection initialization method already provides strong initialization information for Gaussian depth attributes based on depth estimates, and the final Gaussian depth attributes do not exhibit significant errors from the depth estimates.
> > > >
> > > > Similarly, in ablation analysis, we demonstrate results without using the proposed initialization strategy. In such cases, depth information of Gaussian components contains errors, leading to spatial structure corruption where Gaussian ellipsoids floating at incorrect spatial positions cause unintended occlusion of subsequently placed object Gaussians.
> > > > Our proposed image Gaussian representation is an explicit representation of images, and fitting errors affect final rendered image quality. However, existing works using diffusion model-based methods to generate object insertion result images have background (non-target region) preservation capability bounded by the base diffusion model's reconstruction capacity upper limit. Additionally, modifications to the latent of the inserted object region inevitably affect other regions of the original scene image during decoding.
> > > >
> > > > Results reported in our quantitative experiments validate that our proposed image Gaussian representation method has stronger image reconstruction/fitting capability than current state-of-the-art diffusion model-based methods for generating object insertion result images.
> > > >
> > > > Finally, we sincerely thank you again for your contributions and dedication to the community. We remain open to addressing your questions and suggestions.

---

> ### Author Response · Authors · 2025-11-27
> **Follow up**
>
> Dear Reviewer TPis,
>
> We appreciate your time and effort in reading our response and revision! We have now provided detailed responses to all comments and made corresponding revisions where possible. With the discussion period ending around December 3, we would be very grateful if you could kindly take a moment to review our clarifications and let us know if any further information is needed.
>
> Best regards,
>
> Authors

---

> > ### Comment · Reviewer_TPis · 2025-11-28
> > **Response to authors**
> >
> > I thank the authors for their response. As I am not an expert in this specific area, I still have some concerns about the method’s computational efficiency compared to approaches that do not rely on on-the-fly models. I will finalize my recommendation after considering the comments of the other reviewers.

---

> > > ### Author Response · Authors · 2025-11-28
> > > **Further Clarification on the Computational Efficiency of Our Method Compared to Other Approaches**
> > >
> > > Dear Reviewer TPis,
> > >
> > > We sincerely appreciate the time and effort you have dedicated to reviewing our work.
> > >
> > > Since the generative domain entered the era of large-scale foundation models, both train-based and train-free methods mentioned in our paper have been built upon pretrained large models. To achieve superior performance, leveraging stronger pretrained foundation models has become the prevalent approach. Consequently, recent works are predominantly based on Flux, where even inference alone—without any training—requires over 30GB of GPU memory.
> > >
> > > Train-based methods typically introduce trainable modules to accept task-specific conditional inputs when adapting pretrained large models to new tasks. These methods also incorporate parameter-efficient fine-tuning techniques such as LoRA into the backbone network, and train the newly introduced parameters on curated domain-specific datasets for the target task. During inference, these methods perform direct forward passes without requiring additional parameter optimization. The recent methods we mentioned in our previous response, InsertAnything and Unicombine, both belong to this category. Training on large-scale models like Flux to support new tasks demands substantial computational resources and memory (typically requiring multiple 80GB GPUs for parallel training), along with the construction of large-scale domain-specific datasets (usually tens of thousands of image pairs). These requirements impose significant limitations on such approaches.
> > >
> > > In contrast, train-free methods primarily operate by directly manipulating the inference process of pretrained large models or utilizing inference results as supervision to accomplish target tasks, without involving dataset construction or fine-tuning of the foundation model. Our method also operates solely through manipulating the inference process of the pretrained large model and leveraging inference results as supervision. The portions involving large model inference in our entire pipeline take no more than 10 seconds. Furthermore, the process of fitting our proposed image Gaussians itself has extremely low computational requirements (requiring only 2GB of memory for 1024×1024 resolution images).
> > >
> > > Best regards,
> > >
> > > Authors

---

### Official Review · Reviewer_8ePN · 2025-10-29

**Soundness:** 3
**Presentation:** 3
**Contribution:** 3
**Rating:** 6
**Confidence:** 3

**Summary:**

This paper introduces "Spatial Composer," a novel framework for object insertion. The core idea is to represent both the scene and the reference object as "Depth-Aware Image Gaussians (DA-ImgGS)," a tailored variant of 3D Gaussian Splatting (3DGS) optimized for single-image reconstruction. Object insertion is then modeled as a composition of these Gaussian sets in 3D space. This approach elegantly resolves occlusion by leveraging the inherent depth-sorted rendering mechanism of 3DGS. To ensure seamless integration, the framework employs a refinement pipeline based on pre-trained diffusion and illumination models, which cleverly updates only the object's Gaussian parameters to guarantee perfect preservation of the background.

**Strengths:**

True 3D Spatial Controllability and Inherent Occlusion Handling: The paper's primary strength lies in elevating object insertion from a 2D manipulation task to a 3D composition problem. This allows for precise, intuitive control over an object's placement and scale within the scene's depth. Crucially, complex occlusion relationships are handled naturally and correctly by the depth-aware rendering process, a fundamental challenge that most prior works struggle with.

Intelligent and Novel Adaptation of 3D Gaussian Splatting: The authors do not merely apply 3DGS but adapt it effectively for the ill-posed problem of single-view reconstruction. The proposed DA-ImgGS simplifies the representation for this specific task , and the back-projection initialization strategy provides a robust spatial prior from monocular depth estimation, which is critical for achieving a stable and meaningful 3D structure.

**Weaknesses:**

Insufficient Quantitative and Qualitative Results in the Main Paper: Te experimental validation in the main body of the paper is sparse. The quantitative evaluation relies heavily on a user study, and the newly proposed automatic harmony metrics are not benchmarked against prior work, with their definitions relegated to the appendix. Qualitatively, Figure 2 provides only a limited set of examples, forcing the reader to consult the appendix to assess the method's general performance.

Lack of a Defined Interaction Model: The paper shows final results but fails to address how a user would interact with the system to create them. Manually defining 3D coordinates and scaling factors is not a user-friendly process. The work currently stands as a proof-of-concept, but its practical utility is questionable without a well-defined and intuitive user interface for object manipulation in 3D space.

**Questions:**

Same as weakness.

---

> ### Author Response · Authors · 2025-11-19
>
> Dear Reviewer 8ePN,
>
> We sincerely thank you for your diligent review work and for providing valuable questions and suggestions for our work.
>
> **Response to Weakness 1:**
>
> First, we apologize for the inadequate experimental section due to space limitations and structural organization issues. We appreciate your constructive feedback.
>
> Since many experiments were placed in the appendix, the main text experiments appeared weak. We have restructured the paper, especially the experimental section, in the current version to address this issue. You can download the new version of the paper to view the current organization of the experimental section.
>
> Regarding quantitative metrics, previous works used feature similarity between inserted and original objects to measure identity preservation. However, in object insertion, harmonization and occlusion handling inevitably reduce feature similarity between inserted and original object regions. Simple copy-paste would achieve theoretical upper-bound metrics, so we did not adopt them.
> Following your suggestion, we introduce widely recognized metrics in image/video generation/editing: CLIPScore and VQAScore. CLIPScore, used in previous benchmarks, computes cosine similarity between CLIP image embeddings of the inserted object region and CLIP text embeddings of prompts describing object style and category ("XXX style / professional photograph of a XXX"). VQAScore evaluates overall visual quality and text-prompt alignment, providing automatic evaluation better aligned with human subjective assessment, also computed based on the inserted object region and style/category text prompts.
>
> We also moved experiments on other benchmarks to the main text for more comprehensive presentation.
>
> The challenge of automatic metrics aligning with human perception is common in this field. Due to application complexity, finding effective metrics aligned with human evaluation standards is difficult. Therefore, user studies provide complementary evaluation aligned with human subjective visual perception.
>
> For qualitative aspects, we restructured the paper to provide more representative examples in the main text.
>
> **Response to Weakness 2:**
>
> Thank you for your constructive suggestions, which provide valuable guidance for enhancing the usability of our method.
> We will further improve user-friendliness in subsequent work. For the scene-object Gaussian composition component, we will design an interface where users can directly move and scale objects using mouse or keyboard in a 3D coordinate system, based on visualized scene and object point clouds, to complete controllable composition of scene and object Gaussians. We plan to complete this implementation before the end of the rebuttal period and update screenshots of this interactive visualization process in the appendix.
>
> Finally, we sincerely thank you again for your contributions and dedication to the community. We remain open to addressing your questions and suggestions.

---

> > ### Author Response · Authors · 2025-11-22
> >
> > We have implemented a simple yet effective visualization-based interactive framework for scene-object Gaussian composition, which significantly enhances the usability of the composition process. The technical details of this composition method (primarily based on point cloud visualization of scene and object Gaussian means with mouse and keyboard interaction) are provided in the appendix. Additionally, we have substantially reorganized the experimental section. Please refer to the revised version of our manuscript to review these updates.

---

> > > ### Comment · Reviewer_8ePN · 2025-11-23
> > >
> > > Thanks the author's rebuttal.
> > >
> > > I will keep my score.

---

### Official Review · Reviewer_s7nq · 2025-11-02

**Soundness:** 2
**Presentation:** 3
**Contribution:** 2
**Rating:** 4
**Confidence:** 3

**Summary:**

This paper introduces SpatialComposer, a novel, training-free method for 3D-aware object insertion into images. The core idea is to represent both the scene and the object as "Depth-Aware Image Gaussians" (DA-ImgGS), which are simplified 3D Gaussian Splatting representations initialized using monocular depth estimates. Object insertion is then performed by composing these Gaussian representations—translating and scaling the object's Gaussians within the scene's spatial structure. Finally, a refinement stage uses pre-trained diffusion models to harmonize the inserted object's style and lighting with the scene, while only optimizing the object's Gaussian parameters to preserve the original scene.
The authors identify three key challenges in object insertion: preserving object/scene identity, controlling 3D position/scale, and achieving visual harmony. SpatialComposer directly addresses all three. To evaluate their method, they construct a new, high-resolution benchmark dataset (DAOI) featuring complex spatial structures, which they argue is more suitable than existing benchmarks like TF-ICON. Experiments show that SpatialComposer outperforms several state-of-the-art baselines in user studies and achieves competitive results on quantitative metrics.

**Strengths:**

1.	The paper is exceptionally clear. The problem is well-defined, the method is explained step-by-step, and the results are presented comprehensively with both quantitative tables and qualitative figures.
2.	The back-projection initialization strategy for Gaussians is clever and efficient, ensuring spatial coherence without requiring multi-view inputs or extensive training, while maintaining high-quality reconstruction.
3.	Introduction of the DAOI dataset is a valuable contribution, providing a more challenging benchmark with diverse, high-res scenes and objects, which better tests depth-aware capabilities compared to existing low-res datasets like TF-ICON.

**Weaknesses:**

1.	The refinement step is a practical application of existing models (FLUX-Fill-dev, LBM) but does not introduce a novel algorithm. The latent overwriting mechanism is a known technique in diffusion-based editing. The idea of making only the object Gaussians trainable is a good design choice to preserve the scene, but it is more of an engineering decision within an existing framework rather than a core technical contribution.
2.	While training-free, the refinement step depends on pre-trained models like LBM, which could inherit biases or fail in out-of-distribution styles/artistic domains, despite claimed robustness.

**Questions:**

1.	The authors didn't mention the details of compositing the scene gs and object gs.  Does this happen in a 3dgs editing tool like SuperSplat?  If that's the case, the work feels disjointed as the user will need to reconstruct the gs model, manually merge and place the 2 models, and finally run the refinement process.

  The final step includes using two existing models to refine an image, which does not add much novelty to the work.  The second composition step also does not include technical contribution.  Thus the overall technical contribution is quite limited, though the authors deliver a great solution for object insertion task.

2.	If the inserted object takes up a large part of the original image, the refinement process may affect the background part (non-targeted regions) due to the generative capabilities as also mentioned by the authors.  Do you have any idea to address such problems?

---

> ### Author Response · Authors · 2025-11-19
>
> Dear Reviewer s7nq,
>
> We sincerely thank you for your diligent review work and for providing valuable questions and suggestions for our work.
>
> **Response to Weakness 1:**
>
> Training-free methods cannot fine-tune models or add modules, limiting technical approaches. Most training-free editing methods manipulate latent features through weighted fusion, replacement, or attention-based fusion. Our fusion design is intensity and region-controllable, preserving object identity while adjusting low-level information (style, texture, color) based on global context. Additionally, our refinement leverages the observation that diffusion models form high-level semantics early but focus on low-level details in final denoising steps, which are more relevant to harmonization factors like style and illumination. By noising disharmonious objects at appropriate timesteps and manipulating latents during denoising, we achieve harmonization while preserving identity.
>
> We are the first to address precise 3D spatial position and scale control in object insertion, a practical problem unsolved by prior works.
>
> Making only object Gaussians trainable during harmonization is not our core contribution. Our main contribution is the Gaussian representation itself: using a single image and relative depth estimate, we recover scene representation with usable spatial structure and high reconstruction accuracy. This representation enables distinguishable scene/object Gaussians, differentiable rendering, and end-to-end optimization, which supports our harmonization approach.
>
> **Response to Weakness 2:**
>
> Existing training-free image generation methods rely on pretrained model knowledge, so their performance upper bound is limited by the base model. Our claimed robustness is relative to training-based object insertion methods. Training-based approaches introduce new architectures on top of base text-to-image models (e.g., Stable Diffusion, FLUX) and fine-tune on collected paired datasets. Due to the high cost and difficulty of collecting task-specific paired data, these datasets are typically small-scale and limited in scene diversity. Although base models are trained on massive data, fine-tuning on limited task-specific data causes performance degradation on out-of-distribution samples. Our method directly uses the base model without task-specific fine-tuning, thereby maintaining stronger robustness and generalization compared to fine-tuned models.
>
> We acknowledge your point is correct—this is a common limitation for training-free methods. If the task exceeds the base model's capabilities, performance cannot be guaranteed.
>
> Finally, to avoid misunderstanding: we observed that pretrained text-to-image models have limited understanding of illumination, a core factor affecting real scene harmonization. Therefore, we introduce the LBM model, which is specifically trained for real scene illumination harmonization. We clarify that LBM does not address style/artistic domain issues.
>
> In the large model era, most works build upon pretrained models. Training-based methods also fine-tune powerful base models and achieve excellent results with limited data due to base model support—while also being constrained by base model capabilities. We believe this is an objective reality in the current era.
>
> **Response to Question 1:**
>
> Our method does not rely on other software or tools. Our code provides a complete pipeline from Gaussian fitting and Gaussian composition to object Gaussian harmonization.
>
> Both scene and object Gaussian fitting take RGB images and relative depth estimates as input, automatically complete the fitting process, and output 3D Gaussians with optional saving of Gaussian component mean point clouds. For Gaussian composition, based on the visualization of scene and object point clouds in NDC coordinate space from the previous step, users specify object Gaussian translation and scaling to complete the merging. User specification of object position and scale in the scene ensures the application better aligns with user intent.
>
> We will soon develop a more interactive and user-friendly Gaussian composition process, allowing users to perform merging via mouse or keyboard input in a visualization interface, and will reflect this in the appendix of future paper revisions.

---

> > ### Author Response · Authors · 2025-11-19
> >
> > **Response to Question 2:**
> >
> > To the best of our knowledge, we are the first to address precise 3D spatial position and scale control for inserted objects in object insertion tasks, and provide an effective solution based on our representation design.
> >
> > Regarding the novelty of the harmonization component, please see our response to Weakness 1.
> >
> > The composition step is a control mechanism enabled by our representation for precisely controlling 3D spatial position and scale of inserted objects. This composition approach, supported by our pipeline design, achieves precise 3D spatial control that previous object insertion works could not accomplish. It addresses spatial position and scale control in image object insertion from an image-to-space perspective.
> >
> > We believe our technical contribution lies in providing a scene image representation that can be constructed from a single image, maintains high reconstruction accuracy, and supports extensions for attributes like depth and semantics.
> > Leveraging this representation and existing generative models, we provide a potential application on a currently difficult downstream task without additional model training, achieving comparable or superior results to baseline methods and solving a practical challenge. The object insertion pipeline construction extensively utilizes the properties enabled by our proposed representation.
> >
> > **Response to Question 3:**
> >
> > In fact, our method does not face this issue. While the diffusion model's limitations may cause changes to regions outside the refinement area (i.e., background or non-target regions as mentioned), after the diffusion model produces harmonization predictions, even if the background is modified, our Gaussian-to-image rendering and loss calculation process is fully differentiable. When only object Gaussians are set as optimizable, since object Gaussians do not contribute to background rendering, gradient information from the modified background loss is not propagated to object Gaussians, while background Gaussians remain unchanged.
> >
> > Finally, we sincerely thank you again for your contributions and dedication to the community. We remain open to addressing your questions and suggestions.

---

> > > ### Author Response · Authors · 2025-11-22
> > >
> > > We have provided detailed explanations of the scene-object Gaussian composition method in the appendix. Additionally, we have substantially reorganized the experimental section based on your feedback. We kindly invite you to download the revised version of our paper to review these updates.

---

> ### Author Response · Authors · 2025-11-27
> **Follow up**
>
> Dear Reviewer s7nq,
>
> We appreciate your time and effort in reading our response and revision! We have now provided detailed responses to all comments and made corresponding revisions where possible. With the discussion period ending around December 3, we would be very grateful if you could kindly take a moment to review our clarifications and let us know if any further information is needed.
>
> Best regards,
>
> Authors

---

### Author Response · Authors · 2025-12-03
**General Response to Area Chair**

Dear Area Chair,

We sincerely thank all reviewers for their constructive feedback. Below, we summarize the main concerns raised and our responses, highlighting the key revisions made to our manuscript.

**1 Summary of Reviewer Feedback**

All three reviewers acknowledge that our work addresses a practical and meaningful problem—precise 3D spatial position and scale control in object insertion—and delivers an effective solution with competitive or superior results compared to existing methods. The main concerns center on: $\textbf{(1)}$ the level of technical novelty, $\textbf{(2)}$ experimental presentation, $\textbf{(3)}$ usability and interaction design, and $\textbf{(4)}$ robustness analysis.

**2 Key Contributions Clarified**

We emphasize that $\textbf{we are the first to address precise 3D spatial position and scale control for inserted objects}$ in object insertion tasks. Our core contribution is not the individual component (diffusion refinement) but rather $\textbf{the novel depth-aware image Gaussian representation}$ that enables: $\textbf{(a)}$ scene reconstruction from a single image with high fidelity, $\textbf{(b)}$ distinguishable scene/object Gaussians supporting precise 3D composition, $\textbf{(c)}$ differentiable rendering for end-to-end optimization, and $\textbf{(d)}$ training-free harmonization that preserves background regions through selective gradient propagation.

Regarding the "training-free" terminology questioned by Reviewer TPis, we clarify that this follows established conventions in the image generation/editing community, and provide references to papers that use the same terminology to support our claim.

**3 Major Revisions in Response to Feedback**

**Experimental Section Restructured (Reviewers 8ePN):** We substantially reorganized the experimental section to present more comprehensive quantitative and qualitative results in the main text. We introduce widely recognized metrics (CLIPScore, VQAScore) alongside our user study to provide more robust evaluation.

**Interactive Composition Interface Implemented (Reviewers s7nq & 8ePN):** We developed a visualization-based interactive framework allowing users to manipulate object Gaussians via mouse and keyboard in a 3D coordinate system, significantly enhancing practical usability. Technical details and screenshots are included in the appendix.

**Pipeline Efficiency Documented (Reviewer TPis):** Complete pipeline execution takes 2-3 minutes on an A6000 GPU (or $\sim$1 minute with pre-fitted Gaussians). Memory usage is under 35GB with FLUX-Fill or under 12GB with SDXL-inpainting. For reference, InsertAnything—which achieved second-best results across multiple dimensions and metrics in our experiments—requires 2 minutes to process object insertion on a 1024×1024 resolution background image, with GPU memory consumption reaching 44GB. Our method only requires approximately 10 seconds of pretrained large model (FLUX-Fill) inference to construct supervision signals throughout the entire pipeline. The remaining time is spent on image Gaussian fitting, which has relatively low computational cost and memory requirements (under 2GB GPU memory for a 1024×1024 image).

**Robustness Discussion Expanded (Reviewers TPis):** We clarify that our dataset covers diverse scenarios (9+ object categories, multiple artistic styles, indoor/outdoor settings, various lighting conditions). Our training-free approach inherently offers stronger generalization than fine-tuned methods (Due to the limitations of task-specific fine-tuning data compared to pretraining data), though it is bounded by base model capabilities for challenging physics-based effects.

**4 Addressing the Novelty Concern**

Although the methodology of our refinement process shares similar concepts with existing training-free image editing works (i.e., manipulating intermediate features during model inference), this is due to the inherent constraints of the operational space available to training-free methods. The vast majority of such training-free editing approaches manipulate or perturb intermediate features according to their task objectives to achieve editing goals—this guiding principle is common across training-free image editing works. Our more significant contribution lies in proposing an image representation that can be efficiently fitted from a single image and imprecise depth estimates provided by existing methods, yet achieves usable spatial structure and sufficiently high reconstruction accuracy. Leveraging the properties and advantages offered by this representation, we address a previously unsolvable problem in the object insertion: precise control over the 3D spatial position and scale of inserted objects within image scenes. We provide an effective solution to this practically meaningful problem without requiring additional data for model fine-tuning, while also demonstrating a potential downstream application for our proposed representation.

Best regards,

Authors

---

### Meta-Review · Area_Chair_m4rW · 2026-01-08

**Summary:**

The paper offers limited technical novelty, functioning largely as an engineering integration of existing components, Gaussian Splatting and pre-trained diffusion models, rather than introducing a substantive algorithmic contribution. Reviewers (s7nq and TPis) emphasized that the 'train-free' refinement critically depends on strong base models (e.g., FLUX, LBM), hence inheriting their limitations in handling complex physical effects (e.g., reflections) and out-of-distribution generalization. Reviewers also noted that the overall pipeline appears fragmented, requiring nontrivial manual coordination between Gaussian composition and diffusion-based refinement. Finally, despite arguments for efficiency, concerns remained about the computational and memory overhead of the proposed pipeline relative to simpler and more direct baselines.

**Reviewer Concerns:**

The rebuttal successfully addressed presentation flaws by restructuring experiments, moving results to the main text, and adding standard metrics like CLIPScore and VQAScore. The authors also resolved the 'interaction model' deficit by implementing a new visualization-based interface for object manipulation and clarified the train-free terminology. However, the core concern regarding technical novelty remains outstanding. The method is still viewed as a practical application of existing tools rather than a novel contribution. Despite providing runtime stats, the skepticism regarding computational efficiency (memory usage vs. performance) was not fully alleviated for a Reviewer (TPis). The limitation of handling complex lighting/reflections also remains unresolved.

**Reviewer Scores:**

Reviewer s7nq would have maintained their score of 4. The issue focused on the 'limited technical contribution' and the 'disjointed' nature of the pipeline. While the new interface helped, it did not change their fundamental view that the method is largely an engineering decision within existing frameworks.

Reviewer 8ePN explicitly stated he/she would keep the score of 6. Acknowledged the author's rebuttal and the new interactive interface but decided these improvements were not sufficient to raise the score further.

Reviewer TPis would have maintained the score of 4. In the final comment, the reviewer explicitly noted "still have some concerns about the method's computational efficiency" despite the authors' data. Admitting the lack of expertise, the reviewer would not have moved up due to  the novelty concerns raised by others.

---

### Decision · Program_Chairs · 2026-01-26

Reject